# Patients with Metachronous Peritoneal Metastatic Mucinous Colorectal Adenocarcinoma Benefit More from Cytoreductive Surgery (CRS) and Hyperthermic Intraperitoneal Chemotherapy (HIPEC) than Their Synchronous Counterparts

**DOI:** 10.3390/cancers14163978

**Published:** 2022-08-17

**Authors:** Zoltan Herold, Miklos Acs, Attila Marcell Szasz, Katalin Olasz, Jana Hussong, Max Mayr, Magdolna Dank, Pompiliu Piso

**Affiliations:** 1Division of Oncology, Department of Internal Medicine and Oncology, Semmelweis University, H-1083 Budapest, Hungary; 2Department of General and Visceral Surgery, Hospital Barmherzige Brüder, D-93049 Regensburg, Germany; 3Faculty of Medicine, Semmelweis University, H-1085 Budapest, Hungary; 4Faculty of Medicine, University of Saarland, D-66421 Homburg, Germany

**Keywords:** colorectal neoplasms, cytoreductive surgery, HIPEC, mucinous carcinoma, adenocarcinoma

## Abstract

**Simple Summary:**

Mucinous adenocarcinoma develops in 10–20% of all colorectal cancer (CRC) cases. This subtype is characterized by its worse clinicopathological features, including but not limited to more advanced stages at the time of tumor diagnosis, it being more frequent in the proximal colon, it showing mutation more frequently in CRC-specific protooncogenes, and its impaired response rate to various oncological treatments. Although several studies have investigated the benefits of cytoreductive surgery with hyperthermic intraperitoneal chemotherapy (CRS + HIPEC) in peritoneal metastatic CRC, limited data exist on the effect of mucinous CRC. Therefore, a retrospective study was conducted, and the following novel results were obtained. CRS + HIPEC is advantageous for both mucinous and non-mucinous CRC in metachronous cases, but the same cannot be said for those patients with synchronous metastases: the survival of mucinous CRC patients with synchronous peritoneal metastases was significantly worse despite the use of CRS + HPEC treatment.

**Abstract:**

Background: Mucinous adenocarcinoma is a frequent subtype in colorectal cancer (CRC). A higher initial T-stage, poorer differentiation, worse response to anti-tumor therapies, and shorter survival are characteristic of mucinous CRC. Moreover, the therapeutic benefit of cytoreductive surgery with hyperthermic intraperitoneal chemotherapy (CRS + HIPEC) in mucinous CRC has not been significantly investigated. Methods: A retrospective analysis of 218 CRC patients with synchronous or metachronous peritoneal metastases was conducted. Results: 129 and 89 patients had synchronous and metachronous metastases, and 36 (27.8%) and 22 (24.8%) of these were mucinous CRC, respectively. Mucinous CRC was more frequent in the proximal colon, with a higher T-stage and N-stage and with an average peritoneal carcinomatosis index that was 2 values higher. Disease-specific survival was significantly worse in the synchronous mucinous group (median survival: 22.4 months vs. 36.3 months, *p* = 0.0229). In contrast, no such difference was observed in the metachronous cohort (32.6 months vs. 34.4 months, *p* = 0.6490). Conclusions: In the case of synchronous peritoneal metastases originating from mucinous CRC, the positive effect of CRS+HIPEC cannot be verified, and the added value of this highly invasive treatment is therefore somewhat questioned. However, CRS + HIPEC is recommended for metachronous metastases, since no difference between the two CRC-subtypes could be verified.

## 1. Introduction

As colorectal carcinoma (CRC) is one of the leading types of cancers occurring in the adult population [1], we gain progressively more detailed information about that malignancy. Parallel with the developments regarding comprehensive and targeted treatments in terms of neo/adjuvant chemotherapeutic agents and immunotherapy, sufferers of peritoneal metastases (PM) from a colorectal origin obtain improved perspectives. In this context, an aggressive multimodal treatment with cytoreductive surgery (CRS) and hyperthermic intraperitoneal chemotherapy (HIPEC) has been developed to prolong survival in CRC patients with PM.

Synchronous PM (at the time of treatment for primary cancer) appears in around 8% to 15% of CRC patients, while metachronous PM (during follow-up after primary cancer surgery) appears in a further 10–20% [2]. Although after this multimodal treatment, long-term survival can be achieved in well-selected patients, which was investigated by several studies [3,4,5], to date, the four randomized controlled trials have produced contradictory results regarding the effectiveness of surgical cytoreduction with or without HIPEC for patients with PM of CRC [6,7,8,9]. Thus, there is not yet sufficient evidence to determine whether this approach provides a definitive survival benefit and which patients or patient group benefit best from this approach. Accordingly, the National Comprehensive Cancer Network does not consider CRS and HIPEC as a standard treatment for CRC with peritoneal metastases at present [10,11]. In addition, the studies conducted to date have paid little attention to tumor characteristics and histological subtypes in the context of multimodal therapy. As a result, mucinous colorectal adenocarcinoma, containing 10%–20% of CRC cases [12], is an understudied group of CRC patients.

Therefore, our study elaborates on the survival outcomes of patients with synchronous and metachronous PM, with a special emphasis on mucinous adenocarcinoma treated with intraoperative HIPEC after CRS, which is an uncharted comparison in the relevant literature. Here, we aimed to reveal the differences between the specific variety of mucinous colorectal adenocarcinoma (MAC) and non-mucinous adenocarcinoma (AC) survival rate outcomes in both synchronous and metachronous settings.

## 2. Materials and Methods

### 2.1. Patients and Study Design

This study was performed according to STROBE requirements [13]. From January 2011 to December 2021, a total of 129 and 89 consecutively treated patients with synchronous and metachronous PM originating from histopathologically proven CRC who underwent CRS and HIPEC were reviewed and included. The clinicopathological data had been prospectively entered in the national HIPEC registry administered by the German Society for General and Visceral Surgery (DGAV) and were retrospectively analyzed for this study. All the patients had agreed to data recording in the registry and to the use of their anonymized data for quality assurance and research purposes by written and verbal informed consent prior to surgery. Therefore, and due to the retrospective nature of this study, no institutional or further review board approval was necessary. All patients were treated according to multidisciplinary recommendations.

### 2.2. Details of CRS + HIPEC

A closed HIPEC with a goal temperature of 42 °C with bidirectional HIPEC with oxaliplatin (300 mg/m^2^), fluorouracil (400 mg/m^2^), and folinic acid (20 mg/m^2^) or intraperitoneal chemotherapy with mitomycin C (30 mg/m^2^ of body surface area) was administered immediately after CRS for 30, 60, or 90 min of duration. The HIPEC compound changed from oxaliplatin to mitomycin in 2018 due to an institutional protocol change according to the proven better safety of mitomycin C [14]. Oxaliplatin or mitomycin C was added to a 3000-to-4000 mL isotonic saline solution in accordance with the body surface area of the patients. The mean flow rate was 1400–1800 mL/minute. During HIPEC treatments, temperatures were monitored in the right subphrenic and pelvic area.

The extent of peritoneal dissemination was assessed preoperatively using abdominal and chest CT scans. Clinicopathological data were obtained from prospectively collected database and electronic medical reports. All the patients were staged based on the American Joint Committee on Cancer (AJCC) TNM system [15,16]. Right-sided and left-sided tumors were defined as originating from the cecum, ascending colon, and proximal two-thirds of the transverse colon and from the distal one-third of the transverse colon, descending, and sigmoid colon, respectively [17]. Rectal cancers were investigated separately [18]. Protooncogenes: BRAF, RAS, and Microsatellite Instability (MSI) statuses were obtained, if available. Due to the large number of possible combinations, the chemotherapeutic treatment of patients was recorded as the lineage number of the final treatment the patient received prior to CRS + HIPEC, and the usage of biological agents (anti-VEGF or anti-EGFR recombinant chimeric monoclonal antibody) was also recorded.

The completeness of cytoreduction (CC) was scored as proposed by Sugarbaker: CC-0: no residual disease; CC-1: residual nodules measuring less than 2.5 mm; CC-2: residual nodules measuring between 2.5 mm and 2.5 cm; and CC-3: residual nodules greater than 2.5 cm [19]. In the case of simultaneous liver metastasis adjacent to PM, no more than three liver metastases were considered resectable. The extent of peritoneal disease was assessed by using the peritoneal carcinomatosis index (PCI), which ranges from 1 to 39 [19]. After an individual decision in each case, patients with low-risk disease (younger age, nodal-negative, predicted low PCI) underwent upfront surgery, while high-risk patients (nodal-positive, predicted high PCI, dissemination and metastases in other intraabdominal sites) underwent an initial period of systematic chemotherapy prior to surgery.

### 2.3. Clinical Characteristics

Several variables (pre-surgical, surgical, and postoperative features) were analyzed. Postoperative adverse events were categorized according to the Clavien–Dindo Classification, and a major complication was defined as Grade ≥ III [20]. The macroscopic completeness of the resection was assessed intraoperatively by the surgeon. The postoperative attainment of no evidence of disease (NED) was not performed rutinously as part of the postoperative management in our visceral surgery department, as it is not part of the protocol. This is usually performed by CT and/or by PET CT at the discretion of the oncologists providing further treatment. Disease-specific (DSS) and overall survival (OS) were calculated from the date of surgery (CRS + HIPEC) to the date of cancer-related death or death from any cause, respectively. For non-deceased patients, the time interval between the surgery and the last follow-up date was chosen. The follow-up of patients was terminated on 31 March 2022.

### 2.4. Statistical Analysis

Statistical analysis was performed with R version 4.2.1 (R Core Team, 2022, Vienna, Austria). Welch’s *t*-test, the Wilcoxon rank sum test, and Fisher’s exact test were used for group comparisons between cohorts. Comparisons between multiple groups were performed using ANOVA with Tukey post hoc tests and Kruskal–Wallis tests with Dunn post hoc tests. Patient survival was evaluated using competing risk Cox regression models (R package “survival”, Therneau and Grambsch, version 3.3-1). Three separate endpoint events were defined for DSS survival analyses: death related to cancer and to postoperative complications and lost-to-follow-up (LFU). *p* < 0.05 was considered statistically significant, and *p*-values were corrected with the Holm method [21] for the multiple-comparisons problem. Continuous, count, and survival data were expressed as the mean ± standard deviation, the number of observations (percentage), and the hazard ratio (HR) with a 95% confidence interval (95% CI), respectively. Survival curves were drawn with the R-package “survminer” (Kassambara, Kosinski and Biecek, version 0.4.9).

## 3. Results

A total of 218 CRC patients with PM were included in the study. Study participants were grouped based on when the PM was diagnosed and on tumor histology. A total of 93, 36, 67, and 22 patients were enrolled into the synchronous PM + non-mucinous adenocarcinoma (sAC), synchronous PM + mucinous adenocarcinoma (sMAC), metachronous PM + non-mucinous adenocarcinoma (mAC), and metachronous PM + mucinous adenocarcinoma (mMAC) groups, respectively. The results are presented as follows. (1) The comparison of the synchronous and metachronous cohorts was first performed to ensure that there is no significant difference or confounding between the two cohorts apart from some obvious differences, such as the time between PM diagnosis and HIPEC. (2) AC and MAC histologies were then compared within the synchronous and metachronous cohorts separately. (3) The direct comparison of the four study groups was performed.

### 3.1. Comparisions between the Synchronous and Metachronous Metastasis Groups

A total of 129 and 89 patients belonged to the synchronous and metachronous PM cohorts, respectively. As expected, the time between the diagnosis of the tumor and the CRS + HIPEC treatment was significant. It was approximately 2.5 times higher in the metachronous cohort (median: 7 vs. 18.3 months, *p* < 0.0001) compared to that in the synchronous cohort. A lower initial TNM stage (stage T: *p* < 0.0001; stage N: *p* = 0.0588) and left-sided tumors (34% vs. 45%, *p* = 0.0606) were more common for the metachronous group. Furthermore, the number of patients receiving a higher chemotherapy lineage was also higher (9% vs. 28%, *p* = 0.0005) in patients developing PM metachronously, but the usage of anti-VEGF/anti-EGFR monoclonal antibodies combined with cytotoxic doublets was more common in those patients with synchronous peritoneal metastases (29% vs. 45%, crude *p* = 0.0170). Primary tumor resection was basically performed for every patient in the metachronous study group, while it was only performed in approximately two-thirds of the synchronous metastasis patients (*p* = 0.0004). No further differences in the other parameters examined could be verified between the two cohorts.

### 3.2. Comparisions of AC and MAC Histology within the Synchronous Metastasis Group

A total of 36 (27.8%) and 93 (72.2%) of the 129 synchronous PM CRC patients were assigned to the sAC and sMAC groups, respectively. The pre-, peri-, and postoperative clinical characteristics of the two groups and the whole cohort are summarized in Table 1 and Appendix A.

A comparison of the two synchronous PM groups revealed that, in the patients in the sMAC group, the size of the tumor is more commonly larger (crude *p* = 0.0518), and the distribution of lymph node metastases also differs: stage N0 and N2 or above occurs more often than N1 (crude *p* = 0.0343). The tumor of the cecum and the ascendent colon was marginally more frequent in the sMAC group (43% vs. 61%, *p* = 0.0784). Patients receiving no chemotherapy prior to CRS + HIPEC was more common among the sMAC patients (crude *p* = 0.0201); furthermore, pervious partial peritonectomy was performed less often (crude *p* = 0.0114) in the sMAC patients. A few perioperative differences were also found. PCI was higher in the sMAC group (crude *p* = 0.0080). Peritonectomy of the omental bursa (crude *p* = 0.0281), right-upper quadrant (crude *p* = 0.0840), and left-upper quadrant (crude *p* = 0.0025) and splenectomy (crude *p* = 0.0142) were needed more often in the sMAC group during the CRS + HIPEC procedure (Table 1).

A total of 36 and 6 patients were alive at the end of our observation period, 53 and 25 underwent cancer death, and 3 and 5 LFU events occurred in the sAC and sMAC histology groups, respectively. One postoperative death event was registered within the sAC group due to pulmonary embolism. The 1-year, 3-year, and 5-year survival rates were 88.7%, 47.9%, and 21.5% and 66.3%, 17.1%, and 9.1% in the sAC and sMAC groups, respectively. Both DSS (HR: 1.8620, 95% CI: 1.0900–3.1800, *p* = 0.0229) and OS (HR: 1.9970, 95% CI: 1.2690–3.1450, *p* = 0.0028) were significantly worse in those patients with sMAC (Figure 1). The significant effect of histology could not be eliminated if the baseline hazards of the models were adjusted for either the two HIPEC medications (DSS: *p* = 0.0439; OS: *p* = 0.0064), the duration of HIPEC (DSS: *p* = 0.0472; OS: *p* = 0.0097), the number of chemotherapy lineages (DSS: *p* = 0.0097; OS: *p* = 0.0021), the T-stage (DSS: *p* = 0.0175; OS: *p* = 0.0035), the N-stage (DSS: *p* = 0.0402; OS: *p* = 0.0092), the sidedness (DSS: *p* = 0.0240; OS: *p* = 0.0049), the number of previous incomplete surgeries (DSS: *p* = 0.0279; OS: *p* = 0.0044), or for the presence of other, non-peritoneal metastases (DSS: *p* = 0.0221; OS: *p* = 0.0043).

Due to the late closing date of the patient recruitment period, a significant part of the population might have had an insufficient follow-up time; therefore, it was investigated whether the removal of some patients from the cohort or the adjustment for the year that the CRS + HIPEC procedure was performed affect survival results. Basically, the same results were obtained, even if the patients from 2021 (*n* of removed patients: 13; modified survival models: DSS *p* = 0.0226 and OS *p* = 0.0022) and from 2020–2021 (*n* of removed patients: 23; modified survival models: DSS *p* = 0.0287 and OS *p* = 0.0028) were removed from the cohort for the survival analyses. Similarly, no change in the results occurred (DSS: *p* = 0.0109; OS: *p* = 0.0084) if the original survival model was adjusted for the year that the CRS + HIPEC procedure was performed (Appendix A).

The effect of tumor histology over survival was further investigated in a multivariate setting as well. The parameter selection for the model was based on the literature data and medical/clinical importance: anamnestic, histopathological, surgical, and treatment data was used. The significant univariate effect of histology (sAC vs. sMAC) was eliminated in the multivariate models both for DSS (HR: 1.1150, 95% CI: 0.6760–1.8400, *p* = 0.6695) and OS (HR: 1.3686, 95% CI: 0.7516–2.4920, *p* = 0.3049) by the other characteristics of the disease/patients (Table 2).

### 3.3. Comparisions of AC and MAC Histology within the Metachronous Metastasis Group

A mucinous tumor was found in 22 (24.8%) of the 89 metachronous patients. Similar to those of the synchronous metastasis patients, cecal (7.5% vs. 22.2%, crude *p* = 0.0025) and right-sided tumors in general (32.8% vs. 54.5%, crude *p* = 0.0399) were more common in the mMAC group. Furthermore, peritonectomy of the right-upper quadrant (54.5% vs. 19.4%, crude *p* = 0.0026) and left-upper quadrant (27.3% vs. 7.5%, crude *p* = 0.0238) was needed more often in the mMAC group during the CRS + HIPEC procedure, which is similar to the observation of the sMAC population. It has to be mentioned that, although it was not statistically different, the PCI values were tendentiously higher in the mMAC group (mAC: 7.43 ± 5.74; mMAC: 9.19 ± 5.49; crude *p* = 0.1230). The clinicopathological characteristics of the mAC, mMAC, and the whole metachronous cohort are summarized in Table 3 and Appendix A.

A total of 21 and 7 patients were alive at the end of our observation period, 40 and 14 underwent cancer death, and 6 and 1 LFU events occurred in the mAC and mMAC groups, respectively. No postoperative death was registered. The 1-year, 3-year, and 5-year survival rates were 80.4%, 43.1%, and 23.1% and 72.7%, 43.6%, and 24.9% in the mAC and mMAC groups, respectively. Neither DSS (HR: 0.9250, 95% CI: 0.4715–1.8150, *p* = 0.8210) nor OS (HR: 0.8721, 95% CI: 0.4839–1.5720, *p* = 0.6490) differed between the two histology groups (Figure 2). The same was obtained for all baseline hazard-adjusted models, and, similar to that of the synchronous cohort, no significant effect of tumor histology over survival was found in the multivariate models (DSS: HR: 1.5610, 95% CI: 0.8768–2.7800, *p* = 0.1302; OS: HR: 1.2867, 95% CI: 0.5749–2.8800, *p* = 0.5397; Table 4).

### 3.4. Comparisons of All Four Study Groups

The comparisons between all four study groups were also performed. In addition to the differences presented above, we were able to further verify slight but significant differences between the groups in the T stage (mAC vs. sAC: *p* = 0.0006; mAC vs. sMAC: *p* <0.0001; mMAC vs. sMAC: *p* = 0.0116), the N stage (mAC vs. sAC: *p* = 0.0476), the sidedness of the tumor (mAC vs. sMAC: *p* = 0.0159), the number of performed colostomies (mAC vs. sAC: *p* = 0.0120), the peritonectomies of the right- (mMAC vs. sAC: *p* = 0.0282) and left-upper quadrant (mAC vs. sMAC: *p* = 0.0025) during surgery, the number of any previous surgeries (mAC vs. sAC: *p* = 0.0328; mAC vs. sMAC: *p* = 0.0085), the primary tumor resections (mAC vs. sAC: *p* = 0.0027; mAC vs. sMAC: *p* = 0.0003), and the partial CRSs (mAC vs. sAC: *p* = 0.0024). The *p*-values obtained during these comparisons can be found in Appendix A.

By comparing the survival of the four study groups, we found that the two non-mucinous groups had basically the same survival. In contrast, those patients within the sMAC group had the most inferior survival of all of the four study groups, while mMAC was very similar to synchronous MAC in the first two years of the observation period. In the remainder of the observation period, the survival of those within the mMAC group was more comparable to that of the two non-mucinous groups (Figure 3). The *p*-values for the between-group comparisons of OS and DSS are shown in Table 5. In addition to those presented above in Section 3.2 and Section 3.3, no additional results could be verified with multivariate survival models.

## 4. Discussion

Adenocarcinoma is the most common histological subtype of CRC, approximately 10–20% of which can be characterized as MAC [22], but some geographical differences can be observed in the prevalence of the disease: Western countries are more frequently affected [23,24]. By definition, MAC is described as a unique subtype of AC in which more than 50% of the tumor tissue is comprised of extracellular mucinous components [12,22,24]. It has been previously reported that MAC tends to develop more often in younger patients [25]; furthermore, a higher T-stage and N-stage at the time of diagnosis and the dominance of a right-sided location of the tumor are more characteristic of MAC [12,22]. In the current study, we also further strengthened the right-sided dominance and more advanced stages of colorectal MAC neoplasms, and although the age and sex of the MAC patients did not differ statistically, the tendency to a younger age and to more female patients could be observed. The observations that PCI was higher in MAC patients and that some surgical procedures were more frequently necessary during the CRS in the MAC cohort of patients—namely, the peritonectomy of the omental bursa, right-upper quadrant, and left upper-quadrant and splenectomy—are related to the following with a very high probability. It is known that some of the mucin components (MUC2) are highly associated with enhanced progression and metastatic spread [26,27,28]. The mucinous component of MAC neoplasms may act similarly as the intraperitoneal fluid containing free cancer cells of other non-MAC tumors, ultimately causing a more advanced disease. Moreover, it has been reported that PM is more frequent in MAC [29,30].

Various molecular changes in MAC have also been reported. In a meta-analysis, Reynolds et al. [31] have found that MAC is positively associated with KRAS and BRAF mutations, microsatellite instability, the CpG island methylator phenotype, and altered p53 expression more frequently than those of non-mucinous subtypes. In the current study, the protooncogene testing of study participants could not be evaluated, unfortunately, due to the low availability of these data in our database. Based on the limited number of RAS results we could obtain (RAS mutant rate: AC 45%, MAC 75%), we would most likely have presented the same data as above. In addition, the gene expression profiling of MAC revealed numerous differentially expressed genes involved in cellular differentiation and mucin metabolism (MUC1, MUC2, and MUC5AC) [22,24,25,26,32].

The neo/adjuvant treatment of MAC neoplasms is challenging: an impaired response to treatment and a shorter survival of patients is reported in most of the studies [22,24,33]. Some reports have suggested that the response to treatment in lower-stage MAC patients does not differ from that of non-MAC patients [34,35], but basically all of the studies agree that, in advanced stages, the therapy response of MAC is lower regardless the chemotherapy combination used [22,24,33]. Several novel therapeutic options have also emerged lately, including immunotherapies [22,24], drugs targeting mucins [36], and the possible use of nanoparticle drugs [22]. Clinical studies have investigated the effect of immune checkpoint markers in CRC patients with tumors displaying high microsatellite instability and mismatch repair deficiency [37,38,39,40,41,42]. Anti-PD-1 (programmed cell death protein 1)/anti-PD-L1 (programmed cell death ligand 1) and anti-CTLA4 (cytotoxic T-lymphocyte-associated protein 4) drugs showed the most efficacy [37,38,39,40,41,42], but the mucinous components and their effect were only investigated in the study of Kim et al. [39]. They have found that the number of PD-L1-positive tumor cells was associated with a decreased extracellular mucin amount and that patients lacking the mucinous component had a better response to the anti-PD-L1 treatment [39].

Although the increased occurrence of PM is known for MAC CRC [29,30], and CRS + HIPEC is a viable option to effectively treat peritoneal disseminations, to our knowledge, no study has investigated the effect of MAC in separated synchronous and metachronous settings over CRS + HIPEC so far. The following is known about the relationship between CRC and CRS + HIPEC in general. Judicious patient selection is crucial to selecting eligible patients for whom macroscopic complete cytoreduction seems achievable. Previously, and in the current study, the PCI and completeness of cytoreduction score (CCR) have been identified as predictive for survival after CRS and HIPEC [43]. Likewise, the indication for CRS + HIPEC should be critically evaluated in cases of poorly differentiated tumors or proven lymph node metastases [44]. With regard to histology, the detection of signet ring cells worsens the prognosis to the extent that the median survival is just over one year despite HIPEC and thus represents a contraindication for CRS + HIPEC [45]. Nonetheless, the current literature is lacking a similar mucinous adenocarcinoma-specific study; the present study aims to fill this gap. In a recent study by Dietz et al. [46], the survival outcomes after CRS + HIPEC in synchronous versus metachronous PM of CRC have been compared. Consistent with the available results, they found that synchronous PM patients presented with higher TN-stages (*p* < 0.001) and that mucinous adenocarcinomas were more common in this group (*p* = 0.001). Furthermore, OS was significantly shorter for synchronous PM compared to metachronous PM patients (28 versus 33 months, respectively, *p* = 0.045) [46]. We can draw the same conclusion in this regard—that synchronous peritoneal metastatic disease demonstrates poor tumor characteristics and a more advanced disease. Accordingly, there is an urgent need to further optimize patient selection, finding the right timing of multimodal treatment for different patient groups.

Apart from the above, the prognosis of MAC as to non-MAC is still debatable [22]. Many studies support the worse survival of MAC patients when the subjects are limited to stage III and stage IV diseases [47,48,49]. Huang et al. [50] have already analyzed the survival difference between peritoneal metastatic MAC vs. non-MAC colorectal cancer following CRS + HIPEC. There was no significant difference in OS and DFS between the non-MAC and MAC groups (*p* = 0.657 and *p* = 0.938, respectively) [50]. However, all patients with peritoneal metastasis, regardless of onset (synchronous or metachronous), were analyzed together, making the comparison with the current detailed study difficult. In addition to the study of Huang et al. [50], we can report the following novel results: no difference in patient survival was found if the PM developed metachronously; however, if the PM was found synchronously at the time of tumor diagnosis, those patients with MAC CRC had significantly shorter survival times, and the median survival was more than 1 year shorter (22 months vs. 36 months). It was investigated whether any confounding effect could cause this observation, but neither the more advanced stages, the short adjuvant chemotherapy before surgery, the duration and the medication used during HIPEC, the sidedness, nor the CC score of patients affected the strong effect of MAC vs. AC.

Examining the causes and explanation of poorer survival in mucinous carcinoma, we found several mechanisms. First, mucinous tissue gains easy access in the whole abdominal cavity, causing higher PCI and leading to a poor survival rate [51]. Second, MAC causes larger primary lesions and higher rates of nodal and distant metastasis [47]. Third, the lower responsiveness to chemotherapy, which has been demonstrated by previous reports, means that the mucinous histology generally predicts a reduced response to a 5-FU-, oxaliplatin-, and irinotecan-based regimen [22]. Another important issue is the administration of neo/adjuvant systemic therapy in addition to CRS + HIPEC, as there is currently no consensus regarding the neoadjuvant chemotherapy prior to CRS + HIPEC. The value of perioperative systemic therapy is currently being investigated in the CAIRO-6 trial [52].

### Limitations of the Study

There are some limitations to this study which should be addressed. First, the combination of the relatively small sample size and the usage of *p*-value adjustments caused a loss in significance in the case of several parameters, even in those that were reported in previous studies as well. In order to maintain methodological accuracy, they were interpreted ultimately as clinical trends. Second, due to its retrospective nature, the present study may contain several biases. Some of the participants enrolled in the study were referred to our tertiary center, often with a more complex disease course and extensive peritoneal carcinomatosis. It was reductively followed from the former that the prior neo/adjuvant chemotherapy was administered by other centers, on which we had no influence. The lack of a control group precludes a reliable conclusion on the benefit of HIPEC. Third, the inclusion period of the study was long, which introduced additional heterogeneity due to the fact that several changes in the routine treatment of CRC patients occurred (e.g., RAS/BRAF/MSI is currently a routine procedure, while it was not in 2011). Furthermore, the current study is a 10-year retrospective series, where the duration and medication of HIPEC has changed; however, all other variables, including the surgical team, have remained unchanged.

## 5. Conclusions

In summary, a retrospective study with the inclusion of patients with synchronous or metachronous PM originating from CRC was conducted. The two cohorts were further divided into patients with MAC and AC. In line with previous findings, the dominance of more advanced stages, the location to the proximal colon, and higher PCI scores were found in the MAC patients. Survival analyses revealed that, while there is no difference in patients with metachronous PM, those patients with synchronous PM originating from MAC had worse chances for longer survival.

In conclusion, we found that metachronous colorectal cancer sufferers with mucinous peritoneal metastasis may benefit significantly more from multimodal therapy with CRS + HIPEC than their synchronous counterparts. As a result, our study stands as a preliminary benchmark regarding patient-group selection. Thus, we suggest emphasizing and considering that recommended patient category as the target of intended treatment receivers. Moreover, we highlight the need for further elaborations and randomized trials and encourage the field’s thorough exploration.

## Figures and Tables

**Figure 1 cancers-14-03978-f001:**
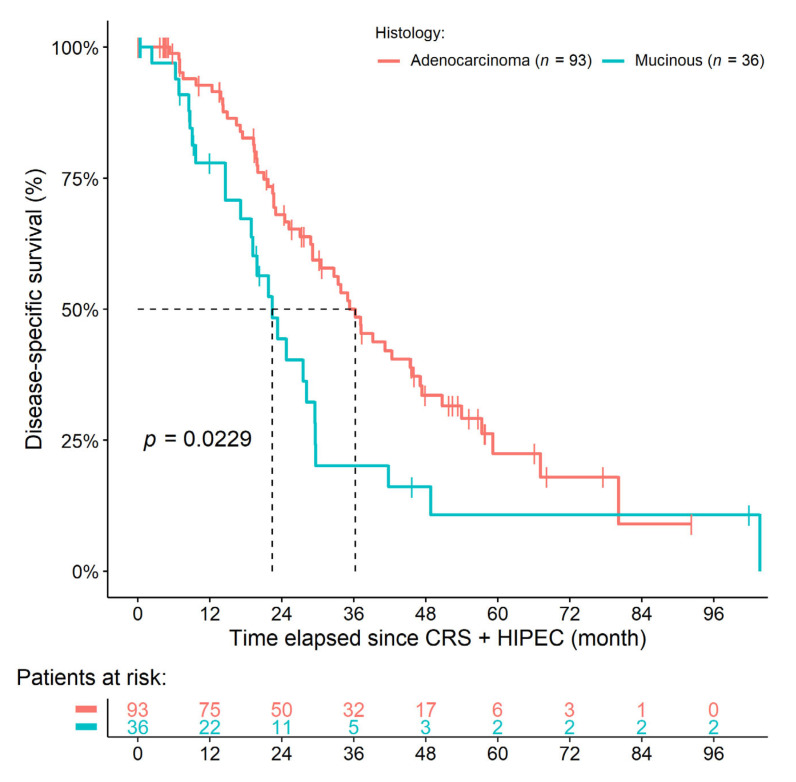
Survival curves of colorectal cancer patients with synchronous peritoneal metastasis. The histological diagnosis of mucinous adenocarcinoma was associated with inferior survival.

**Figure 2 cancers-14-03978-f002:**
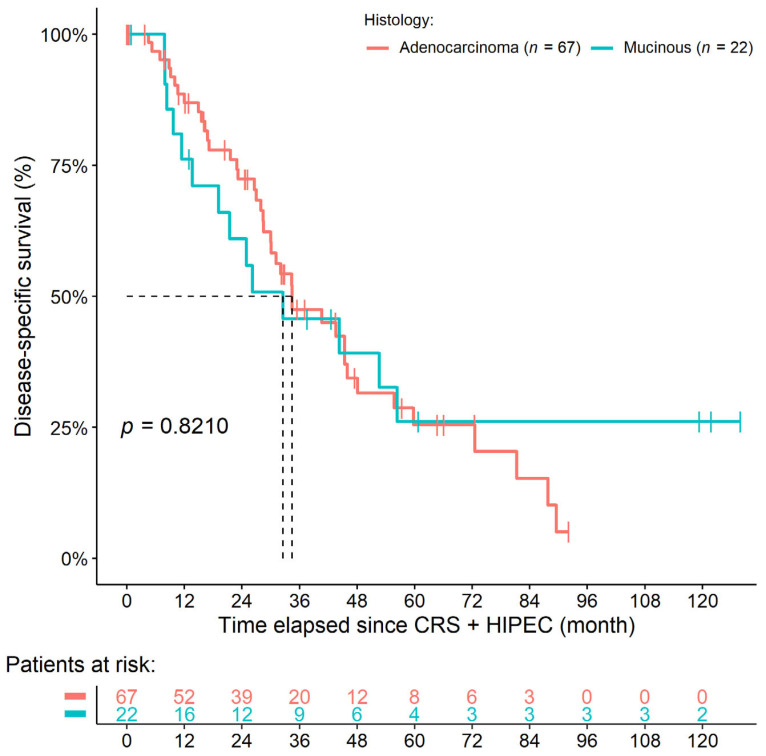
Survival curves of colorectal cancer patients with metachronous peritoneal metastasis. No difference could be found between those patients with mucinous or non-mucinous adenocarcinoma.

**Figure 3 cancers-14-03978-f003:**
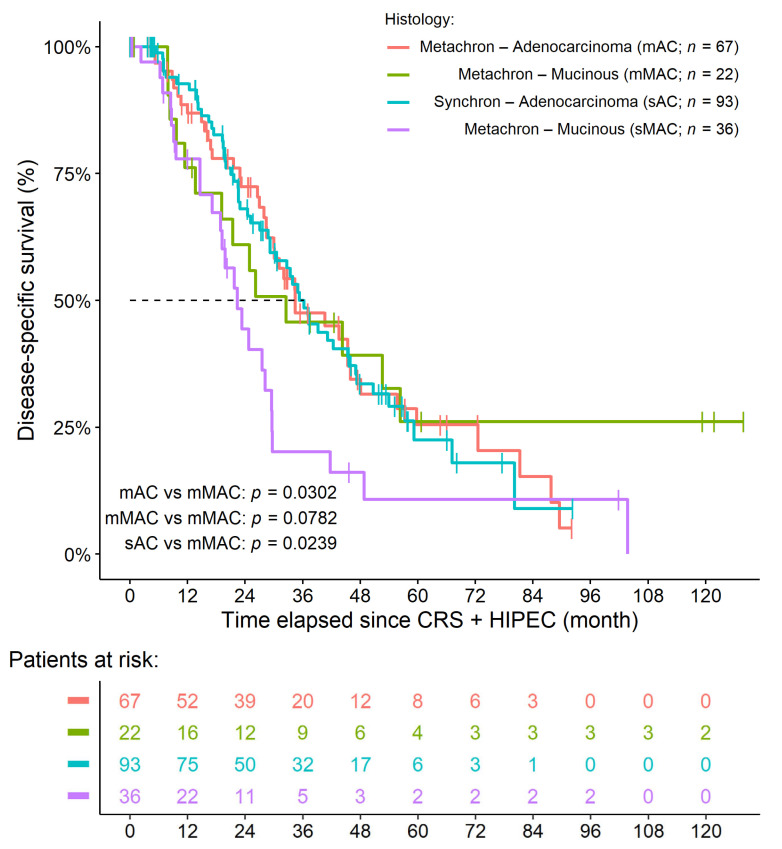
Survival curves of colorectal cancer patients with synchronous and metachronous peritoneal metastasis. The histological diagnosis of mucinous adenocarcinoma in the synchronous cohort was associated with inferior survival compared to all other study cohorts. No statistical difference could be found for the comparisons between the remaining groups.

**Table 1 cancers-14-03978-t001:** Pre-, peri-, and postoperative demographic and clinical characteristics of study participants with synchronous peritoneal metastases (PM). Unit of frequency and survival data are the number of observations (percentage) and the median survival time (95% confidence interval), respectively. Further parameters are presented in Appendix A.

Clinical Characteristics	Synchronous PM (*n* = 129)	sAC (*n* = 93)	sMAC (*n* = 36)	Crude *p*-Value	Adjusted *p*-Value
Stage T (size of tumor) ^1^				0.0518	1.0000
1–2	3 (2.3%)	3 (3.2%)	0 (0%)		
3	31 (24.0%)	27 (29.0%)	4 (11.1%)		
4	93 (72.1%)	62 (66.7%)	31 (86.1%)		
Stage N (lymph node metastases) ^1^				0.0343	1.0000
0	23 (17.8%)	14 (15.1%)	9 (25.0%)		
1	33 (25.6%)	29 (31.2%)	4 (11.1%)		
2–3	71 (55.0%)	48 (51.6%)	23 (63.9%)		
Lineage of chemotherapy				0.0201	1.0000
First line	90 (69.8%)	70 (75.3%)	20 (55.6%)		
Second line	8 (6.2%)	7 (7.5%)	1 (2.8%)		
Third line	3 (2.3%)	1 (1.1%)	2 (5.6%)		
Previous partial peritonectomy ^2^	30 (23.3%)	27 (29.0%)	3 (8.3%)	0.0114	0.6959
Peritoneal carcinomatosis index	7.67 ± 6.63	6.69 ± 5.83	10.22 ± 7.89	0.0080	0.4983
Surgical procedures:					
Perit.: omental bursa	10 (7.8%)	4 (4.3%)	6 (16.7%)	0.0281	1.0000
Perit.: right-upper quadrant	38 (29.5%)	23 (24.7%)	15 (41.7%)	0.0840	1.0000
Perit.: left-upper quadrant	21 (16.3%)	9 (9.7%)	12 (33.3%)	0.0025	0.1585
Splenectomy	9 (7.0%)	3 (3.2%)	6 (16.7%)	0.0142	0.8531
Median overall survival (month)	29.17 (22.97–35.32)	34.99 (28.84–45.67)	19.88 (14.59–28.16)	–	0.0028 ^3^
Median DSS (month)	29.67 (25.20–37.19)	36.27 (29.17–47.08)	22.41 (18.92–29.60)	–	0.0229 ^3^

^1^ TNM information was missing for the 1-1 patient of both groups. ^2^ Including primary tumor removal surgeries, diagnostic/explorative laparoscopies, and metastasectomies. ^3^ *p*-value acquired from the Cox regression survival model. DSS: disease-specific survival; Perit: peritonectomy; sAC: synchronous PM + non-mucinous adenocarcinoma; sMAC: synchronous PM + mucinous adenocarcinoma.

**Table 2 cancers-14-03978-t002:** Results of uni- and multivariate survival models of study participants with synchronous peritoneal metastases.

Clinical Characteristics	*DSS*		*OS*	
	Univariate *p*-Value	Multivariate *p*-Value	Univariate *p*-Value	Multivariate *p*-Value
Age (years)	0.3060	0.0068	0.1460	0.2324
Body mass index (kg/m^2^)	0.7610	0.6984	0.8330	0.9951
Sex (female (ref.) vs. male)	0.2220	0.0005	0.0535	0.0084
ASA score (I-II (ref.) vs. III-IV)	0.9260	0.0850	0.2970	0.4688
CC score				
CC-0 (ref.) vs. CC-1	0.0002	0.8930	0.0008	0.4297
CC-0 (ref.) vs. CC-2	<0.0001	<0.0001	0.0002	0.4779
Peritoneal carcinomatosis index	<0.0001	<0.0001	<0.0001	0.0284
Histopathology (Normal adenocarcinoma (ref.) vs. mucinous)	0.0229	0.6695	0.0028	0.3049
Duration of HIPEC				
30 min (ref.) vs. 60 min	<0.0001	0.0036	0.0058	0.4879
30 min (ref.) vs. 90 min	0.2390	0.3216	0.4234	0.3481
Lineage of chemotherapy				
None (ref.) vs. first-line	0.9550	0.3215	0.8024	0.3830
None (ref.) vs. second-line	0.1600	<0.0001	0.1370	0.4480
None (ref.) vs. third-line	<0.0001	<0.0001	0.0025	0.0012
Usage of biological agents				
None (ref.) vs. anti-VEGF	0.2437	0.1874	0.0437	0.5559
None (ref.) vs. anti-EGFR	0.8621	0.0819	0.7991	0.3321
Any incomplete tumor removal surgery prior to CRS + HIPEC				
None (ref.) vs. single	0.4100	0.2237	0.3570	0.4283
None (ref.) vs. multiple	0.7150	0.0123	0.5670	0.3720
Primary tumor resecated (Yes (ref.) vs. No)	0.1840	0.0594	0.1490	0.4018
Sidedness				
Left-sided (ref.) vs. right-sided	0.0282	<0.0001	0.1040	0.0070
Left-sided (ref.) vs. rectum	0.5587	0.8102	0.3590	0.6583
T stage				
I-II (ref.) vs. III	0.1930	0.0658	0.2250	0.4211
I-II (ref.) vs. IV	0.5270	0.0038	0.5300	0.6362

ASA: American Society for Anesthesiologists; CC: Sugarbaker’s completeness of cytoreduction score; CRS: cytoreductive surgery; DSS: disease-specific survival; HIPEC: hyperthermic intraperitoneal chemotherapy; OS: overall survival.

**Table 3 cancers-14-03978-t003:** Pre-, peri-, and postoperative demographic and clinical characteristics of study participants with metachronous peritoneal metastases (PM). Unit of frequency and survival data are the number of observations (percentage) and median survival time (95% confidence interval), respectively. Further parameters are presented in Appendix A.

Clinical Characteristics	Metachronous PM (*n* = 89)	mAC (*n* = 67)	mMAC (*n* = 22)	Crude *p*-Value	Adjusted *p*-Value
Sidedness				0.0734	1.0000
Left-sided	45 (50.6%)	33 (49.3%)	7 (31.8%)		
Right-sided	29 (32.6%)	22 (32.8%)	12 (54.5%)		
Rectum	14 (15.7%)	12 (17.9%)	2 (9.1%)		
Unknown primary location	1 (1.1%)	0 (0%)	1 (4.5%)		
Surgical procedures:					
Perit.: right-upper quadrant	25 (28.1%)	13 (19.4%)	12 (54.5%)	0.0026	0.1616
Perit.: left-upper quadrant	11 (12.4%)	5 (7.5%)	6 (27.3%)	0.0238	1.0000
Median overall survival (month)	32.10 (26.97–45.40)	32.10 (27.93–45.44)	26.18 (19.12–NA) ^1^	–	0.8210 ^2^
Median DSS (month)	34.43 (28.48–48.13)	34.43 (29.96–48.13)	32.56 (19.12–NA) ^1^	–	0.6490 ^2^

^1^ Not enough observation to reach the lower 95% confidence interval. ^2^
*p*-value acquired from the Cox regression survival model. DSS: disease-specific survival; mAC: metachronous PM + non-mucinous adenocarcinoma; mMAC: metachronous PM + mucinous adenocarcinoma; Perit: peritonectomy.

**Table 4 cancers-14-03978-t004:** Results of uni- and multivariate survival models of study participants with metachronous peritoneal metastases.

Clinical Characteristics	*DSS*		*OS*	
	Univariate *p*-Value	Multivariate *p*-Value	Univariate *p*-Value	Multivariate *p*-Value
Age (years)	0.9941	0.9141	0.4630	0.2623
Body mass index (kg/m^2^)	0.2740	0.4648	0.2010	0.4721
Sex (female (ref.) vs. male)	0.8615	0.5201	0.6160	0.1225
ASA score (I-II (ref.) vs. III-IV)	0.6940	0.2470	0.5690	0.2725
CC score				
CC-0 (ref.) vs. CC-1	0.7410	0.3935	0.9980	0.5185
CC-0 (ref.) vs. CC-2	<0.0001	0.6079	0.3210	0.4917
Peritoneal carcinomatosis index	0.0606	<0.0001	0.0564	0.0034
Histopathology [normal adenocarcinoma (ref.) vs. mucinous]	0.8210	0.1302	0.6490	0.5397
Duration of HIPEC				
30 min (ref.) vs. 60 min	0.5760	0.2094	0.3310	0.1302
30 min (ref.) vs. 90 min	0.1750	0.0617	0.2270	0.3711
Lineage of chemotherapy				
None (ref.) vs. first-line	0.2782	0.7994	0.2787	0.3441
None (ref.) vs. second-line	0.0909	0.2421	0.0885	0.1340
Usage of biological agents				
None (ref.) vs. anti-VEGF	0.3177	0.2857	0.3720	0.2599
None (ref.) vs. anti-EGFR	0.0673	0.2533	0.2910	0.7365
Any incomplete tumor removal surgery prior to CRS + HIPEC				
None (ref.) vs. single	0.0634	0.0001	0.4550	0.3465
None (ref.) vs. multiple	0.3192	<0.0001	0.6290	0.2752
Primary tumor resecated (Yes (ref.) vs. No)	0.3550	<0.0001	0.3400	0.1621
Sidedness				
Left-sided (ref.) vs. right-sided	0.6200	0.0270	0.7080	0.4033
Left-sided (ref.) vs. rectum	0.7290	0.6344	0.5800	0.9616
T stage				
I-II (ref.) vs. III	<0.0001	<0.0001	0.8670	0.8323
I-II (ref.) vs. IV	<0.0001	<0.0001	0.9840	0.8258

ASA: American Society for Anesthesiologists; CC: Sugarbaker’s completeness of cytoreduction score; CRS: cytoreductive surgery; DSS: disease-specific survival; HIPEC: hyperthermic intraperitoneal chemotherapy; OS: overall survival.

**Table 5 cancers-14-03978-t005:** Comparison of the survival of the four study groups. The orange and blue sections show the disease-specific and overall survival between-group *p*-values, respectively.

Histological Subtypes According Occurrence of Peritoneal Metastases	mAC	mMAC	sAC	sMAC
**mAC**	–	0.7728	0.9686	0.302
**mMAC**	0.5966	–	0.7873	0.0782
**sAC**	0.6899	0.7861	–	0.0239
**sMAC**	0.118	0.0179	0.0031	–

mAC: metachronous peritoneal metastasis + non-mucinous adenocarcinoma; mMAC: metachronous peritoneal metastasis + mucinous adenocarcinoma; sAC: synchronous peritoneal metastasis + non-mucinous adenocarcinoma; sMAC: synchronous peritoneal metastasis + mucinous adenocarcinoma.

## Data Availability

The data presented in this study are available on request from the corresponding author.

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
