# Peer review of "Patients with Metachronous Peritoneal Metastatic Mucinous Colorectal Adenocarcinoma Benefit More from Cytoreductive Surgery (CRS) and Hyperthermic Intraperitoneal Chemotherapy (HIPEC) than Their Synchronous Counterparts"

_cancers, 2022, doi:10.3390/cancers14163978_

Round 1

Reviewer 1 Report

This is an interesting retrospective study, looking at the effect of CRS-HIPEC in patients with CRC according to the histological subtype (mucinous vs not) and timing of the occurrence of metastases (synchronous vs metachronous). The authors found that patients with mucinous histology have less benefit from CRS-HIPEC in synchronous disease. No difference is found in metachronous disease.

The findings are interesting and clinically relevant, whereas based on the retrospective nature of the analysis and the mentioned limitations not practice changing. Despite of that it is worth being published in cancers after consideration of some minor issues.

Minor comments:

Methods

Inclusion period was from Jan 2011 to Dec 2021: what was the median follow up time? Including patients until Dec 2021 is not unproblematic for analyses like this, especially in patients with a median OS around 3 years. It would be better to include patients until Dec 20 (or 2019).

Results

Table 1 contains a bunch of variable, which are obviously not relevant for the results (f.e. detailed complications, duration of HIPEC etc.). On the other hand, the patient characteristics of the metachronous cohort is banned to the supplementary data. It would be more comprehensive to shorten the table 1 containing variables that are necessary for understanding the following results including the metachronous cohort (details for understanding “no difference” in OS are necessary as well). Details of the patient characteristics could then be added to the supplementaries.

Table 1: could you explain how patients with “synchronous metastatic disease” can be stage I-III? Aren’t all of them stage IV? Size of primary tumor (as mentioned) does not include stage IV on the other hand.

Did the patients had metastases outside of the peritoneum as well? From the operation procedures I think so – did this have an influence (tumor burden, site of metastases) on outcome?

As you have so many clinical variables available it would be interesting to see a multivariate analysis and if mucinous histology really is the dominant prognostic factor (maybe within another project). Have you looked at this (stepwise exclusion for example)?

Did all patients receive neoadjuvant and/or adjuvant treatment? Did this influence outcome?

You nicely discuss the chemo-sensitivity troubles of mucinous CRC: could you show the response rate to eventual neoadjuvant treatment and OS according to response rate for both, mucinous and non-mucinous?

It appears, that all patients reached no evidence of disease (NED) after multimodal treatment (including metastasis resection outside of the peritoneum)– is that correct? What was the time point OS was calculated from – was it from CRS-HIPEC or from first time point of NED? Or was CRS-HIPEC always the last step before reaching NED? Reason for the question is, that ~35 months OS is good, but could be expected higher in such a patient group (31 – 38 months in systemic treatment trials like CALGB80405, FIRE3/4 etc.).

Discussion

Well dicussed

Limitations

Inclusion of patients from 2011 to 2021 is a limitation in my opinion, since several changes in treatment (RAS-triggering, left vs right sided disease…) have been taken place.

Reviewer 2 Report

Authors present an interesting retrospective assessment of a very niche area. The main conclusion is CRS and HIPEC benefits those with metachronous mucinous colorectal peritoneal metastasis over synchronous. I feel the data could have presented in a more user friendly manner to draw this observation out. I do not feel that authors directly compare synchronous and metachronus groups very clearly and would advise a figure segregating the disease simply into synchronous mucinous/adeno v metachronous mucinous/adeno (4 groups) and make a comparison among these groups. The way the data is presented does not back this main conclusion. Additionally this is a bold statement given the group of mucinous metachronous patients numbers 22. Indeed it is more nuanced than this as mucinous neoplasms do no better than adenocarcinoma in the case series presented when metachronous. I think direct comparison is necessary to draw these conclusions and improved presentation of data in this regard may allow this work that is limited by small sample size and biases author's have  identified to be published.  

Round 2

Reviewer 2 Report

I think the additional figure helps the paper, however, I feel the data presentation of the paper could be more logical and clearer in general and this has not been considered in such as short time frame. 
